# ONNX-Net: Towards Universal Representations and Instant Performance Prediction for Neural Architectures

## Abstract

Neural architecture search (NAS) automates the design process of high-performing architectures, but remains bottlenecked by expensive performance evaluation. Most existing studies that achieve faster evaluation are mostly tied to cell-based search spaces and graph encodings tailored to those individual search spaces, limiting their flexibility and scalability when applied to more expressive search spaces. In this work, we aim to close the gap of individual search space restrictions and search space dependent network representations. We present ONNX-Bench, a benchmark consisting of a collection of neural networks in a unified format based on ONNX files. ONNX-Benchincludes all open-source NAS-bench-based neural networks, resulting in a total size of more than 600k {architecture, accuracy} pairs. This benchmark allows creating a shared neural network representation, ONNX-Net, able to represent any neural architecture using natural language descriptions acting as an input to a performance predictor. This text-based encoding can accommodate arbitrary layer types, operation parameters, and heterogeneous topologies, enabling a single surrogate to generalise across all neural architectures rather than being confined to cell-based search spaces. Experiments show strong zero-shot performance across disparate search spaces using only a small amount of pretraining samples, enabling the unprecedented ability to evaluate any neural network architecture instantly.

## 1 Introduction

Neural Architecture Search (NAS) aims to automate the design of neural networks, with the goal of surpassing manually developed architectures and enabling the discovery of novel network types. However, NAS has largely failed to deliver on its promise of uncovering fundamentally new architectures—such as facilitating the shift from convolutional networks to transformers. One contributing factor is the use of restrictive search spaces, like cell-based search spaces, which limit exploration to a single class of network designs (Ying et al., 2019; Chen et al., 2021). Recently, researchers have begun to focus on more expressive search spaces that enable the discovery of more diverse and innovative architectures (Schrodi et al., 2023; Ericsson et al., 2024).

However, the design of search spaces is only one aspect of NAS. Based on the design, the actual search strategy within the search space is the biggest computational bottleneck, as the number of possible architectures increases significantly with increasing size and expressiveness of the search space.

To improve on the search cost, surrogate models (Dudziak et al., 2020; Lee et al., 2021) were introduced. These surrogate models learn a mapping from architecture representations to their evaluation performance, enabling search methods to explore more candidates per unit time. Early surrogate models (Tang et al., 2020; Wen et al., 2020) typically relied on graph-based encodings of architectures using Graph Neural Networks (GNN) (Kipf & Welling, 2017). While effective within their target settings, these designs are based on strong priors about the search space, such as fixed cell topologies and constrained node counts, making them difficult to transfer across tasks and search spaces.

More recent approaches adopt graph-based representations (Mills et al., 2023; Akhauri & Abdelfattah, 2024) to improve generalisation across search spaces. GENNAPE Mills et al. (2023) is able to represent any neural network using computational graphs encodings of operation bundles as layer information; a $3 \times 3$ convolution operator is a bundle of Conv3x3-BN-ReLU. FLAN Akhauri & Abdelfattah (2024) uses adjacency-matrix encodings. However, these approaches have important limitations. First,

Figure 1: Overview of our approach: ONNX-Bench contains {architecture, accuracy} pairs from multiple search spaces in unified ONNX representation; ONNX-Net consists of a robust, universal text encoding and an LLM-based performance predictor.

adjacency-matrix encodings scale well only in cell-based spaces, where the number of nodes is tightly controlled, and struggle to extend to more flexible search spaces (Ericsson et al., 2024; Schrodi et al., 2023) with sparse variable sized graphs. Second, bundle-based structures mainly capture topology and are largely insensitive to operator parametrisation: two architectures with identical graphs but different hyperparameters (e.g., convolution kernel size, stride, padding or dilation) that share the same graph structure are often indistinguishable. This underlines the strong need for a representation that captures both topology and rich operator-level details in a search space-agnostic manner.

To address this, we develop a more general and robust encoding for NAS surrogate models, **ONNX-Net** (cf. fig. 1 (right)) — one that is agnostic to search spaces, sensitive to operator-level details, and simple to extend. We propose representing architectures as text generated from Open Neural Network Exchange (ONNX) based computational network representations (Bai et al., 2019) and show that training an end-to-end LLM-based predictor on the text encoding allows for instant performance prediction.

Text encodings are flexible and compositional; they can naturally capture topology, operators, and fine-grained parameters, as well as auxiliary context, without redesigning the encoder for each space. However, learning effective predictors from text also requires a sufficiently diverse collection of architectures spanning multiple search spaces. Existing NAS benchmarks fall short in this regard, as they are typically confined to a single network type and cannot support training predictors that generalise beyond it.

To overcome this limitation, we introduce **ONNX-Bench** (cf. fig. 1 (left)), a benchmark that consolidates architectures from multiple search spaces into a unified ONNX-based representation (Bai et al., 2019) with a consistent evaluation setup. ONNX-Bench provides the diversity needed for training and testing cross-space predictors, and serves as a foundation for studying encodings that capture both structural and operator-level details, such as our proposed ONNX-Net. In our experiments, we demonstrate that a surrogate model using the novel text-based encoding trained on ONNX-Bench achieves competitive performance, especially for zero-shot transferability with minimal pretraining.

Our contributions are as follows:

- We release an open-source collection of neural networks in a unified ONNX format, evaluated on CIFAR-10 (Krizhevsky, 2009), enabling research that goes beyond the boundaries of individual search spaces

- We propose a novel ONNX-to-text encoding method that applies to arbitrary architectures and leverages the generalisation ability of large language models

- We present initial experiments on surrogate modeling with this representation, showing strong performance with few pretraining examples and good zero-shot transfer across search spaces

## 2    RELATED WORK

### 2.1    NETWORK SEARCH SPACES

The effectiveness of neural architecture search depends strongly on the design of the underlying search space. Foundational work explored macro-designs using ResNet-style (He et al., 2016) building blocks with skip-connections (Zoph & Le, 2016). Building on this idea, cell-based designs became widely adopted, where a single cell structure is searched and stacked to form the network (Ying et al., 2019; Dong & Yang, 2020; Dong et al., 2021; Liu et al., 2018; Zela et al., 2020). While computationally efficient, cell-based search spaces are much more restrictive, motivating the design of more expressive alternatives. Hierarchical search spaces (Liu et al., 2017; Ru et al., 2020; Schrodi et al., 2023) address this by allowing multi-level structural variations. Beyond hierarchy, recent works focus on topological flexibility and hybridization: Pasunuru & Bansal (2020) relaxes node constraints to capture complex recurrent structures , while Li et al. (2021) and Tỳbl & Neumann (2025) bridge diverse families (e.g., CNNs and Transformers) through fabric-like grids or universal graph decompositions. More recently, grammar-based search spaces were proposed (Schrodi et al., 2023; Ericsson et al., 2024) offering a more flexible and principled way of generating diverse architectures. Notably, `einspace` (Ericsson et al., 2024) overcomes the bias of hand-designed search spaces, using a probabilistic grammar to encapsulate a wide range of architectural families.

Despite these advances, search spaces have been evaluated in isolation due to different design constraints. This highlights the need to unify the search spaces. Ideally, such a framework would combine all {architecture, accuracy} pairs within those search spaces, to enable an unbiased, diverse search space for evaluation of performance predictors. Most previous attempts to create such a search space have used python code as architecture representation (Rahman et al., 2025; Gao et al., 2025; Zhou et al., 2025; Nasir et al., 2024; Chen et al., 2023). While very general, this form of representation comes with its own drawbacks, for example, across different deep learning libraries (e.g., PyTorch, TensorFlow, JAX), the same architecture can be implemented with markedly different code. Even within a single framework, the identical model can be expressed in many syntactically distinct ways (e.g., varying module organization, helper functions, or class structures), creating a many-to-one mapping from Python code to the underlying model.

In this paper, we take a more holistic view by introducing ONNX-Bench, a dataset that unifies multiple search spaces under a common ONNX-based network format. This enables performance prediction and search methods to operate across diverse search spaces within a consistent representation, facilitating fair comparison and more general NAS approaches. Another central advantage of structured file formats such as ONNX is the high expressivity, similar to python code, while yielding far fewer distinct encodings per model. We hypothesize that ONNX representations are substantially more sample-efficient than Python code representations, because the model need not learn invariances to a large variety of superficial code-level differences.

### 2.2    NETWORK ENCODINGS

Orthogonal to the design of the search space, the network encoding plays a crucial role in the search process, especially in order to facilitate speed up techniques such as performance prediction methods. Due to the popularity of cell-based search spaces, most encoding approaches define architectures as graph data and encode them as adjacency matrices processed by Graph Neural Networks (GNNs) (Yan et al., 2020; Ning et al., 2022; Velickovic et al., 2017). Building on these graph representations, Hwang et al. (2024) incorporates information flows within the neural architecture, while Ji et al. (2025a) separates causal and non-causal features of architectures for better prediction. To overcome the pure adjacency based structure, Lukasik et al. (2025) used zero-cost proxies as architecture encodings, and Kadlecová et al. (2024) included additional search space specific network topology information as an input to a tabular prediction method. Recently, Mills et al. (2023); Akhauri & Abdelfattah (2024) learn graph-based encodings with the focus on the ability to transfer between search spaces. While Mills et al. (2023) learns a graph encoder using contrastive learning, Akhauri & Abdelfattah (2024) combines different search-space specific learned encodings, such as an unsupervised learned latent space encoding (Yan et al., 2020), a learned cell encoding using GNNs and zero-cost proxies to learn a network representation. However, these network representations are search space specific, especially with the restriction of only being applicable in cell-based search spaces, eventually limiting the flexibility and scalability of the encoding.

Table 1: Composition of the ONNX-Bench dataset.

| Search Space | Type | Evaluation | Num Architectures |
|---|---|---|---|
| NAS-Bench-101 | | | 423624 |
| NAS-Bench-201 | | CIFAR-10 | 15625 |
| NATS-Bench | Cell-based | | 32768 |
| NAS-Bench-301 | | | 57189 |
| TransNAS-Bench-101 | | Other | 38895 |
| hNAS-Bench-201 | | CIFAR-10 | 8000 |
| einspace | Hierarchical | | 57495 |
| | | UnseenNAS | 16000 |
| Total | | | 649596 |

## 2.3 LLMs in NAS

Recently, LLMs for NAS have become quite a common approach, fuelled by the frequent publication of ever more capable language models. Common modes of operation are LLMs as performance predictors (Jawahar et al., 2023) or for generating/mutating network architectures, often in combination with evolutionary algorithms. Most approaches do apply LLMs in settings where topology or operations are confined by some prior structure (e.g. a cell-based search space (Cai et al., 2025; Zhong et al., 2024; Zheng et al., 2023), supernets (Ji et al., 2025b; Jawahar et al., 2023), choosing parameters only in predefined ranges (Qin et al., 2024), or manipulating sequences, where entries have predefined meanings (Dong et al., 2023; Hu et al., 2025)). Qin et al. (2025) proposed to use a string representation of a network, based on the grammar in einspace, with an LLM for performance prediction to overcome the lack of flexibility, showing improvements over the usage of zero-cost proxies and topology features as in Kadlecová et al. (2024). However, all these methods are dependent on the search space, and cannot be transferred from one search space to another. In this work, we aim to push the boundaries of NAS and present ONNX-Net, a universal network encoding, independent of the search space, with the ability of instant performance prediction. This encoding is independent of the search space design and allows encoding any neural network that can be converted into an ONNX file.

## 3 ONNX-Bench

We introduce a new benchmark dataset for neural architecture search and performance prediction. It collates networks across several sources in a unified format and evaluation setup. This is needed to take NAS beyond the restriction of individual smaller search spaces, and will enable the training of performance predictors that can transfer across existing and future search spaces.

To achieve high compatibility with different frameworks and network formats, we chose the ONNX file format (Bai et al., 2019) as a basis for our work, as it is a de-facto standard for neural network persistence. This allows our method, cf. section 4, to be applied to nearly any network found in the wild, as most frameworks and file formats support saving or conversion into ONNX.

ONNX is a binary file format that represents neural network architectures as directed graphs, where nodes are instances of a set of pre-defined operations, while edges represent tensors/arrays passed between these operations. Nodes also contain the hyperparameter values for their operations, e.g. the kernel size of a pooling operation. Additionally, every ONNX file contains a list of input tensors (and their values, in the case of learned parameters), and output tensors.

The benchmark includes networks from multiple sources, spanning both cell-based and macro-level search spaces such as NAS-Bench-101 (Ying et al., 2019), NAS-Bench-201 (Dong & Yang, 2020), NATS-Bench (Dong et al., 2021), DARTS-style cells in NAS-Bench-302 (Zela et al., 2020), and the hierarchical spaces hNAS-Bench-201 (Schrodi et al., 2023), and einspace (Ericsson et al., 2024). Standardising these diverse architectures into ONNX allows them to be easily compared, reused, and extended. Table 1 summarises the distribution of architectures across sources, along with statistics such as the number of nodes and operator types.

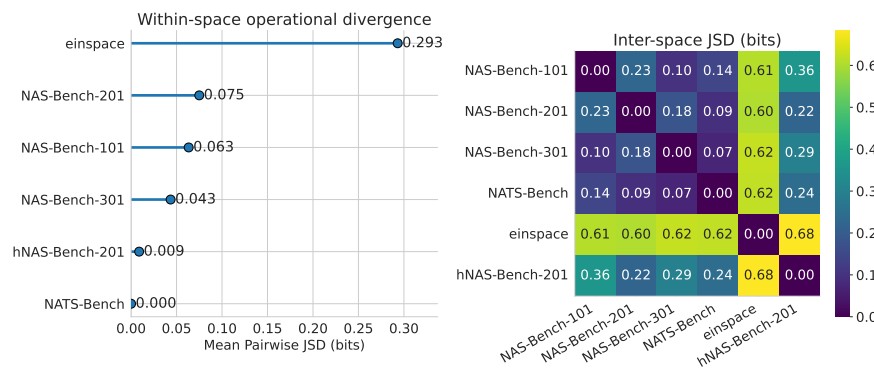

Figure 2: Diversity within and between the search spaces in ONNX-Bench, diversity is measured using Jensen-Shannon divergence (in bits).

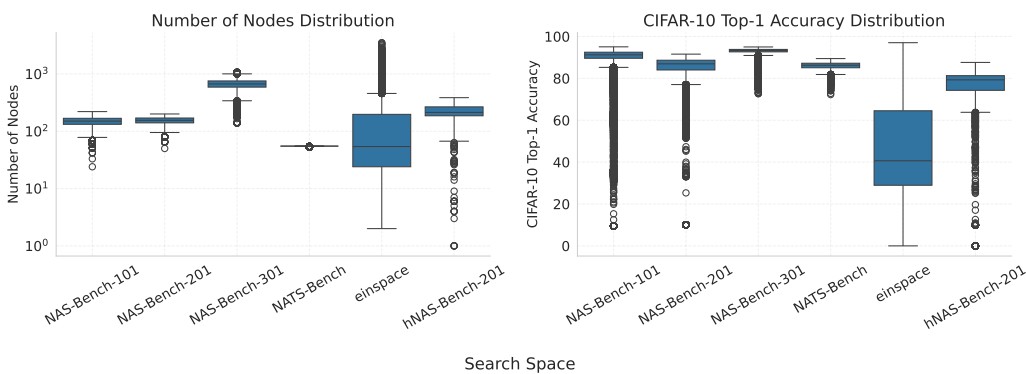

Figure 3: Distribution of nodes (left) in log scale and CIFAR-10 accuracy (right) of the search spaces contained in ONNX-Bench.

All architectures are evaluated on the CIFAR-10 dataset using a consistent training pipeline. CIFAR-10 is widely used in NAS research and provides a balance between computational tractability and benchmark relevance. By fixing the dataset and training settings, ONNX-Bench ensures that observed performance differences reflect the architectures themselves rather than inconsistencies in training protocols.

In total, ONNX-Bench comprises 649596 trained models, with node counts ranging from 1 to 3503 and CIFAR-10 accuracies spanning [0.0, 97.03]. This diversity includes both poorly performing and competitive architectures, making the benchmark suitable for evaluating predictors across the full performance spectrum. We expect ONNX-Bench to support the development of NAS methods that transfer across search spaces and reduce the need for repeated, costly retraining of architectures from scratch.

To further analyse the difference between each search space, we calculate diversity metrics both within and between search spaces in ONNX-Bench. For each model, we count ONNX `node.op_type` occurrences (excluding Constant) and normalize to a probability $p$ over the op vocabulary $V$. For a sampled set $S$ of $n$ models from a search space. We compute Jensen–Shannon divergence (JSD, base-2, in bits) between all pairs:

$$JSD(p_i, p_j) = \frac{1}{2}KL(p_i \| m) + \frac{1}{2}KL(p_j \| m), \quad \text{where } m = \frac{1}{2}(p_i + p_j) \tag{1}$$

In terms of across-space dissimilarity, for two spaces $A$ and $B$, we pool $n$ sampled models in each space to obtain $p_{pool}^A$ and $p_{pool}^B$ over joint vocabulary $V = V_A \cup V_B$, then report $JSD(p_{pool}^A, p_{pool}^B)$ in bits.

The ideal NAS search space encompasses all well-performing neural network architectures possible. While this is infeasible to achieve, we argue that a very diverse search space comes closest to this goal. In fig. 2,

we report diversity measures over 5k random samples from each search space, and show that hierarchical search spaces, such as einspace and hNAS-Bench-201, differ strongly from other spaces (and, in the case of einspace, also have a much broader variety of architectures). That these additional architectures are not only composed out of low-performing "fail-cases" can be clearly seen in fig. 3. As ONNX-Bench encompasses all aforementioned search spaces, it fulfils the goal of architecture-diversity to a high degree.

# 4 ONNX-NET

As a baseline for future performance predictors developed on ONNX-Bench, we propose a presentation that allows to describe any neural network architecture in the ONNX format and can be easily coupled with an instant performance prediction on a given dataset. Figure 4 shows an overview of our approach.

The central part of our proposed approach is the representation of the neural architecture in the form of natural language. To represent the information contained in the ONNX files as text, we first reconstruct the graph contained in the file in memory, retaining all information. Since context length is a limiting factor for many LLMs, we try to reduce the size of the graph by performing a variety of optimisations:

**Node removal** We remove nodes we deem to be of low importance, such as identity operations or input nodes for parameters, which can be implicitly inferred by their context.

**Subgraph merging** We merge common subgraphs with known interpretation into single nodes, e.g. merging a matrix-vector multiplication with a parameter, followed by an addition with a parameter, into a single node, corresponding to a linear layer.

Some of the (lossless) optimisations were performed by the *ONNX-Simplifier* tool (@daquexian et al., 2019). The resulting, shortened graph is again shrunk by merging chains of operations without any branches, to obtain the final, condensed version of the graph. We then convert this graph structure into text by printing each chain of nodes on one line, in the following format:

```
Operation(Input1, ...)(Parameter1=Value,...) -->
    Operation(prev, ...)(Parameter1=Value,...) -->
    --> ... --> Output1, ..., OutputN:Shape
```

To show the applicability of ONNX-Net we train an LLM to act as an exemplary surrogate model. NAS is prohibitively expensive, due to the need of training every candidate architecture to evaluate its performance. Surrogate models circumvent this reoccurring cost by acting as a performance predictor, inferring the performance of a candidate architecture of some search space (in this case all architectures representable by ONNX) by information obtained without training the candidate. This accelerates the search, as the inference speed of many neural networks is negligible in comparison to the amount of time a (partial) training may require.

Formally, our goal is to create a learned predictor $P_\theta(\cdot) : \mathcal{A} \to \mathbb{R}$, which is capable of ranking any architecture in $\mathcal{A}$ by their performance on a certain dataset in a certain metric, e.g. ranking architectures by their accuracy on CIFAR10. Here, $\mathcal{A}$ denotes the space of all possible ONNX-encoded neural network architectures. We further define $\phi(\cdot;\mathcal{D}):\mathcal{A}\to\mathbb{R}$ as the training process for any network $a\in\mathcal{A}$ on dataset $\mathcal{D}$, which returns the validation performance. Evaluating $\phi$ is computationally expensive, therefore, minimizing the number of evaluations is the primary motivation for creating a surrogate predictor. The parameters $\theta$ of $P_\theta$ are trained on samples of {architecture, accuracy} pairs $\{(a_i,\phi(a_i;\mathcal{D}))\}_{i=1}^N$ by minimizing an empirical risk objective

$$\hat{\theta}=\operatorname*{argmin}_\theta \frac{1}{N}\sum_{i=1}^N \mathcal{L}(P_\theta(a_i),\phi(a_i;\mathcal{D})). \tag{2}$$

$\mathcal{L}$ is a loss function that quantifies the discrepancy between the predicted and true performances. This loss can take the form of e.g. a mean-squared error or a ranking loss.

# 5 EXPERIMENTS

ONNX-Bench can be a valuable dataset for the NAS community to investigate search space-independent surrogate models that generalise to various types of architectures. As a baseline, we evaluate ONNX-Net

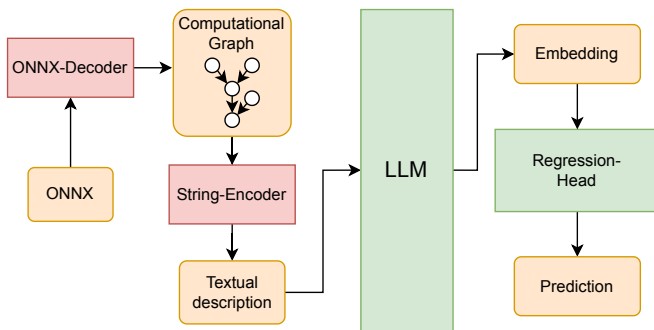

Figure 4: The ONNX-Net pipeline for performance prediction of any ONNX-encoded neural network architecture.

Table 2: Kendall's $\tau$ correlations for surrogate models trained on the full set of search spaces as well as all-but-one search space. Rows after the first show the spaces left out of the training set and columns show the evaluation spaces where we compute correlations on held-out data.

| | | NAS-Bench-101 | NATS-Bench | NAS-Bench-301 | hNAS-Bench-201 | einspace |
|---|---|---|---|---|---|---|
| | Train on all | 0.772 | 0.788 | 0.691 | 0.533 | 0.477 |
| Leave-out | NAS-Bench-101 | 0.529 | 0.815 | 0.687 | 0.386 | **0.529** |
| | NATS-Bench | 0.744 | 0.390 | **0.704** | 0.382 | 0.478 |
| | NAS-Bench-301 | 0.777 | 0.819 | 0.508 | 0.214 | 0.474 |
| | hNAS-Bench-201 | 0.787 | 0.825 | 0.693 | **0.565** | 0.524 |
| | einspace | **0.794** | **0.843** | 0.694 | 0.456 | 0.301 |

with respect to its ability to predict performances on new search spaces (section 5.1), with respect to its zero shot performance (section 5.2), and its ability to generalise to new tasks (section 5.3).

**Metrics** We report the rank correlation values (Kendall's $\tau$, Spearman's $\rho$) between the ground-truth and predicted performances.

**Model** We fine-tune a `ModernBERT-large` model as the performance predictor, and we also compare it with other LLMs in section 6.2.

## 5.1 How well does the surrogate perform on new search spaces?

We access how well surrogate trained on all but one search space generalise to the excluded search space (table 2). For better comparison, we also include surrogate trained on all search space.

Given that NATS-Bench is extension of NAS-Bench-201, they are considered together as NATS-Bench in this experiment. The detailed train/val split regime is described in appendix B.

Results in table 2 shows training on all search spaces yields strong but not uniformly optimal Kendall's $\tau$ across targets. Holding out a space typically degrades performance on that space, with a notable exception for hNAS-Bench-201, where leaving it out actually improves transfer (0.533 → 0.565), suggesting negative transfer when hNAS is included. The best per-column scores are generally achieved without the full mixture, suggesting future work regarding finding the optimal data mixture for a universal surrogate model.

## 5.2 Zero-shot performance across search spaces

To enable comparison with prior work (Akhauri & Abdelfattah, 2024; Mills et al., 2023), we evaluate the zero-shot transfer from `NAS-Bench-101` to `NAS-Bench-201`. Concretely, we train the surrogate on 50k random {architecture, accuracy} pairs from `NAS-Bench-101` and evaluate on the full `NAS-Bench-201` set without any adaptation.

Table 3: Zero-shot predictor trained on 50k samples from NAS-Bench-101 and evaluated on NAS-Bench-201. Avg. Spearman's $\rho$ over 5 random seeds is reported.

| Transfer | GENNAPE | - | CATE | FLAN Arch2Vec | ZCP | CAZ | ONNX-Net |
|---|---|---|---|---|---|---|---|
| Zero-Shot | **0.815** | 0.697 | 0.697 | 0.741 | 0.646 | 0.685 | 0.747 |

Table 4: Zero-shot transfer learning among einspace, NAS-Bench-101 and NAS-Bench-201. Avg. Spearman's $\rho$ over 5 random seeds is reported. We also report Avg. Kendall's $\tau$ in table 10.

| Source → Target Search Space | Train Size | | |
|---|---|---|---|
| | 200 | 1000 | 5000 |
| NAS-Bench-101 → einspace | 0.264 | 0.199 | 0.155 |
| einspace → NAS-Bench-101 | 0.310 | 0.351 | 0.348 |
| einspace → NAS-Bench-201 | 0.242 | 0.198 | 0.180 |

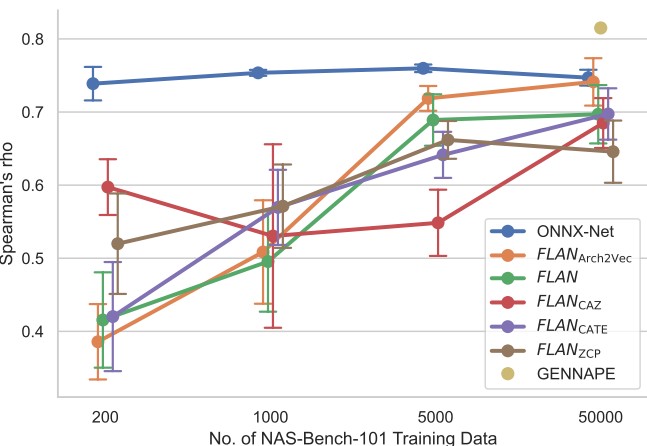

Figure 5: Zero-shot predictor trained on different number of `NAS-Bench-101`, evaluated on NAS-Bench-201. Avg. Spearman's $\rho$ and standard error over 5 random seeds is reported.

We also study data scaling by varying the `NAS-Bench-101` training set size: 200, 1k, and 5k samples. We replicate the FLAN setup using their released code under our protocol; we are unable to replicate GENNAPE due to the lack of reproducible codes and therefore we report only results from their paper.

Results in table 3 and fig. 5 show that GENNAPE achieves the strongest zero-shot transfer overall ($\rho$=0.815), noting that it utilises an ensemble combining multiple predictors with two pairwise classifiers. Relative to FLAN, our surrogate consistently achieves higher zero-shot performance across all training-set sizes, including FLAN variants that incorporate additional encodings such as CATE, Arch2Vec, or zero-cost proxies (ZCP); the gains are largest in the low-data regime. Our surrogate reaches its peak zero-shot correlation with 5k training samples and exhibits substantially lower seed-to-seed variance than FLAN.

According to analysis in fig. 2, Jensen-Shannon divergence between NAS-Bench-101 and NAS-Bench-201 is 0.23, which explains the good zero-shot transfer ability. Additionally, we also explore zero-shot transfer ability across more distinct search spaces. Result in table 4 shows weaker transfer performance when divergence is high. This outcome underscores our motivation for ONNX-Bench: gathering a unified, diverse architecture collection to expose the surrogate to heterogeneous architectural patterns and improve generalization to truly unseen spaces.

## 5.3 HOW WELL DOES THE SURROGATE DO ON NEW DATASETS?

We further assess the ability of the surrogate model to generalise to classification tasks other than CIFAR10.

Table 5: Kendall's $\tau$ correlations for zero-shot Unseen NAS tasks

| | AddNIST | Language | MultNIST | CIFARTile | Gutenberg | Isabella | GeoClassing | Chesseract |
|---|---|---|---|---|---|---|---|---|
| Full | 0.364 | 0.338 | 0.449 | 0.351 | 0.582 | 0.248 | 0.095 | 0.294 |
| w/o einspace | 0.156 | 0.131 | 0.245 | 0.058 | 0.317 | 0.135 | 0.249 | 0.302 |

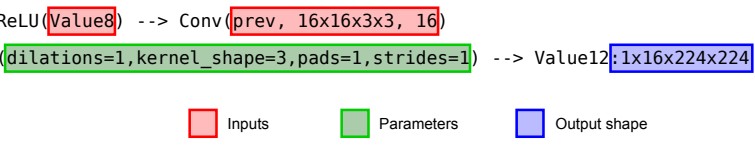

Figure 6: Information to include in text encoding

Table 6: Zero-shot transfer learning between NAS-Bench-101 and NAS-Bench-201 using different text encodings. Avg. Spearman's $\rho$ over 5 random seeds is reported. We also report Avg. Kendall's $\tau$ in table 11.

| Encoding | NB101 → NB201 | | | | NB201 → NB101 | | |
|---|---|---|---|---|---|---|---|
| | 200 | 1000 | 5000 | 50000 | 200 | 1000 | 5000 |
| Base | 0.618 | 0.644 | 0.691 | 0.666 | 0.599 | 0.667 | 0.691 |
| +Inputs | **0.746** | 0.752 | 0.756 | 0.743 | 0.682 | **0.755** | 0.780 |
| +Parameters | 0.726 | 0.724 | 0.715 | 0.718 | 0.713 | 0.720 | 0.762 |
| +Out Shape | 0.660 | 0.679 | 0.693 | 0.682 | 0.658 | 0.686 | 0.675 |
| Full | 0.739 | **0.754** | **0.760** | **0.747** | **0.715** | 0.740 | **0.781** |

**CIFAR-10 to Unseen NAS on einspace (cross-dataset).**    We train a surrogate using CIFAR-10 labels and evaluate zero-shot on eight Unseen NAS (Geada et al., 2024) datasets within the einspace search space. Results in table 5 show that including einspace during training (*Full*) markedly improves zero-shot transfer to UnseenNAS. *Full* outperforms *w/o einspace* on 6 of 8 datasets. Overall, including einspace training data is critical for cross-dataset generalisation within the einspace search space, though the optimal training mix may be task-dependent.

## 6   ABLATION STUDY

### 6.1   TEXT ENCODING

We decompose the architecture-to-text encoding into four components (fig. 6): (i) *Base information*: operation name and output index; (ii) *Input information*: weight/bias shapes and names of the inputs to each operation; (iii) *Parameter information*: operation-specific parameters (e.g., kernel shape and padding for convolutions); (iv) *Output shape information*: the tensor shape of operation's output. To assess the importance of each component, we compare: (a) a *base* variant using only the Base information; (b) the *full* encoding (all four components); (c) three variants where we add one component to the base variant. All models are trained and evaluated under the same setting as section 5.2, we additionally add zero-shot experiments from NAS-Bench-201 to NAS-Bench-101, excluding the 50k variant due to data limitations for NAS-Bench-201.

Table 6 shows that enriching the encoding with *Input information* yields the largest single-step gain over the *Base* variant across both settings, highlighting the importance of explicit connectivity and weight shape cues at the inputs. Adding only *Parameter information* helps when data size is small but offers diminishing gains as the train size grows. *Output shape* alone provides only marginal improvements over *Base*, which matches the expectation as it adds least amount of information. The *Full* version lags slightly behind +Inputs for some cases, likely due to the longer sequences reducing sample efficiency, but best overall. We also observe a mild dip at 50k relative to 5k for most variants, consistent with the observation in section 5.2; this points to potential overfitting to the source domain.

Table 7: Zero-shot transfer learning from NAS-Bench-101 to NAS-Bench-201 using different LM backbone. Avg. Spearman's $\rho$ over 5 random seeds is reported. We also report Avg. Kendall's $\tau$ in table 12.

| Model | Model Size | NB101 $\rightarrow$ NB201 | | | |
|---|---|---|---|---|---|
| | | 200 | 1000 | 5000 | 50000 |
| ModernBERT-base | 150M | 0.725 | 0.730 | 0.744 | 0.737 |
| ModernBERT-large | 396M | **0.739** | **0.754** | **0.760** | **0.747** |
| Qwen3 | 752M | 0.620 | 0.696 | 0.747 | 0.745 |
| Qwen3 | 2.03B | 0.660 | 0.735 | 0.734 | 0.728 |

## 6.2 BASE MODEL CHOICE

We evaluate the effect of the LM backbone by fine-tuning two families: an encoder-based `ModernBERT` and a decoder-based `Qwen3`. For each family, we use the same fine-tuning recipe, data, and evaluation protocol as in section 5.2, results listed in table 7.

Across all data regimes, the encoder-based LM outperforms the decoder-based ones for zero-shot transfer from NAS-Bench-101 to NAS-Bench-201, with `ModernBERT-large` being consistently best. Scaling helps within the encoder family: `ModernBERT-large` surpasses `ModernBERT-base` at every data size. For `Qwen3`, the larger variant is better only when data size is small. The superiority of encoder-based LM matches the observation in Qin et al. (2025).

## 7 CONCLUSION

We have introduced ONNX-Bench, a unified collection of NAS benchmarks containing architectures in a shared ONNX format and performance scores on a common dataset, CIFAR-10. This benchmark can be used to evaluate performance predictors on a general suite of architectural styles, going beyond the existing narrow cell-based benchmarks. In the future we hope to use it for developing general search methods as well.

Using this novel benchmark, we develop and evaluate a novel surrogate model we call ONNX-Net. It uses our condensed string encoding of the ONNX representation as the input to an LLM, fine-tuned towards the performance prediction task. It shows very strong zero-shot performance using only a small amount of training data. Compared to previous methods it can handle more general and flexible architecture inputs, though it also becomes clear that the problem is more difficult and more restricted graph-based approaches can outperform our more generally applicable method. We hope that the release of this benchmark, encoding and surrogate can spur more research into search space-agnostic NAS.

Future work can build upon the benchmark to create search methods in a general architecture format such as ONNX. Continuing in the direction of LLM and string representations can lead to guided generation of architecture candidates. Furthermore, we acknowledge the limitation of this work to mainly focus on performances on the CIFAR-10 dataset, and hope that future work will expand the capabilities of surrogates to take dataset as context. Finally, due to its general scope, we hope to continue expanding ONNX-Bench as a living benchmark, to make it more diverse e.g. with attention-based archiectures.

### REPRODUCIBILITY STATEMENT

To ensure the reproducibility of our work, we submitted anonymous downloadable source code as supplementary materials, along with text encodings used for ONNX-Net experiments and some examples from ONNX-Bench due to the size limit. We also lists hyperparameters used in our experiments in table 9.

### LLM USAGE

Within the scope of this paper, LLMs are used only to aid and polish writing, as well as auto-complete code fragments.

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

## A UNSEEN NAS DATASETS

In the following we will provide an overview of the 8 Unseen NAS tasks (Geada et al., 2024) used in table 5. All tasks are image classification tasks, comprising different difficulties. AddNIST is based on the MNIST dataset (LeCun & Cortes, 2010) using three channels laying in top of each other, with the label being the sum of the label of each channel.The language dataset aims to classify one of 10 languages. The language image is generated by randomly selecting four words from the target language, which are concatenated into a string. This string is then encoded into a $24 \times 24$ grid, where a black pixel indicates the presence of a letter at that grid position. MultNNIST is similar to AddNIST but uses the multiplication of the three channels as a target. CIFARTile is a combination of four CIFAR-10 images in a $2 \times 2$ grid. The target here is the number of distinct CIFAR-10 classes shown in the grid. Gutenberg aims at classifying authors based on three words of consecutive sequences encoded similar to the Language dataset. The Isabella dataset classifies four music eras using recordings of these eras which are converted into 64-band spectrograms. GeoClassing uses patches from the BigEarthNet (Sumbul et al., 2019) dataset to identify the corresponding country. Lastly, Chesseract contains images of the chess boards, of the final $15\%$ of the board state, of public games from eight grandmasters. The target here is to depict the classes: white wins, draw, black wins.

## B SPLITTING METHOD PER SPACE

- NAS-Bench-201 and NATS-Bench: Randomly sample 20% of architectures as validation. Due to the high similarity between these two spaces, we merge them together.
- NAS-Bench-101: We adopt the validation indices provided by the paper Akhauri & Abdelfattah (2024).
- HNAS-Bench-201 and einspace: One search seed is reserved as validation split. For einspace, the validation mirrors the setup in Qin et al. (2025).
- NAS-Bench-301: Three sources are randomly picked as validation split.

Table 8 summarises the number of training and validation instances used from each search space.

Table 8: Dataset sizes per search space. For NAS-Bench-201 and NATS-Bench, we merge their respective train/validation partitions due to space similarity. All labels are CIFAR-10 top-1 accuracy.

| Search space | Train | Validation |
|---|---|---|
| NAS-Bench-101 | 40,000 | 7,290 |
| NAS-Bench-201 + NATS-Bench | 38,714 | 9,679 |
| NAS-Bench-301 | 40,000 | 5,892 |
| hNAS-Bench-201 | 6,403 | 1,000 |
| einspace | 37,416 | 1,582 |

## C IMPLEMENTATION DETAILS

### C.1 RANDOM SEEDS

We use random seed 42 through 46 for all our experiment on multiple seeds. The random seed is used for both training sample selection and training itself.

### C.2 TRAINING HYPERPARAMETERS

The hyperparameters used for training is listed in table 9.

## D GRAPH OPTIMISATIONS

## E STRING ENCODING EXAMPLE

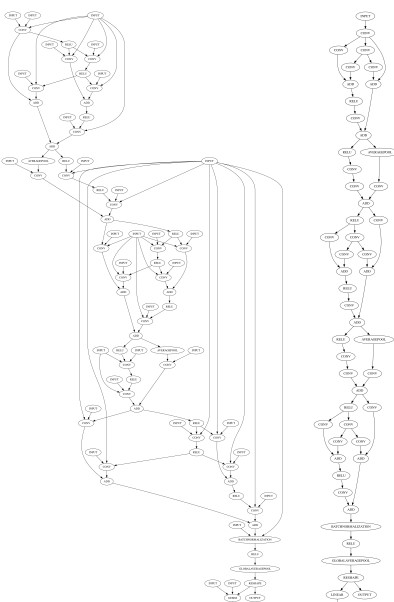

Figure 7: Visualisation of the graph optimisation for a simple neural network. The ONNX graph on the left is simplified in a lossless fashion into the graph on the right.

Table 9: Hyperparameters used in main table experiments.

| Hyperparameter | Value | Hyperparameter | Value |
|---|---|---|---|
| Learning Rate | 5e-5 | Weight Decay | 0.1 |
| Number of Epochs | 5 | Batch Size | 16 |
| Learning Rate Scheduling | Polynomial | End Learning Rate | 5e-6 |
| Gradient Accumulation | 1 | Warm-up Ratio | 0.06 |
| Loss Type | Pairwise Hinge Loss | BF16 | True |

Table 10: Zero-shot transfer learning among einspace, NAS-Bench-101 and NAS-Bench-201. Avg. Kendall's $\tau$ over 5 random seeds is reported.

| Source → Target Search Space | Train Size | | |
|---|---|---|---|
| | 200 | 1000 | 5000 |
| NAS-Bench-101 → einspace | 0.264 | 0.199 | 0.155 |
| einspace → NAS-Bench-101 | 0.310 | 0.351 | 0.348 |
| einspace → NAS-Bench-201 | 0.242 | 0.198 | 0.180 |

Table 11: Zero-shot transfer learning between NAS-Bench-101 and NAS-Bench-201 using different text encodings. Avg. Kendall's $\tau$ over 5 random seeds is reported.

| Encoding | NB101 → NB201 | | | | NB201 → NB101 | | |
|---|---|---|---|---|---|---|---|
| | 200 | 1000 | 5000 | 50000 | 200 | 1000 | 5000 |
| Base | 0.457 | 0.475 | 0.521 | 0.493 | 0.453 | 0.510 | 0.529 |
| +Inputs | **0.566** | _0.571_ | _0.574_ | _0.560_ | 0.509 | **0.577** | _0.606_ |
| +Parameters | 0.549 | 0.542 | 0.536 | 0.536 | **0.540** | 0.543 | 0.581 |
| +Out Shape | 0.496 | 0.509 | 0.523 | 0.511 | 0.499 | 0.524 | 0.513 |
| Full | **0.566** | **0.574** | **0.581** | **0.571** | _0.539_ | _0.564_ | **0.608** |

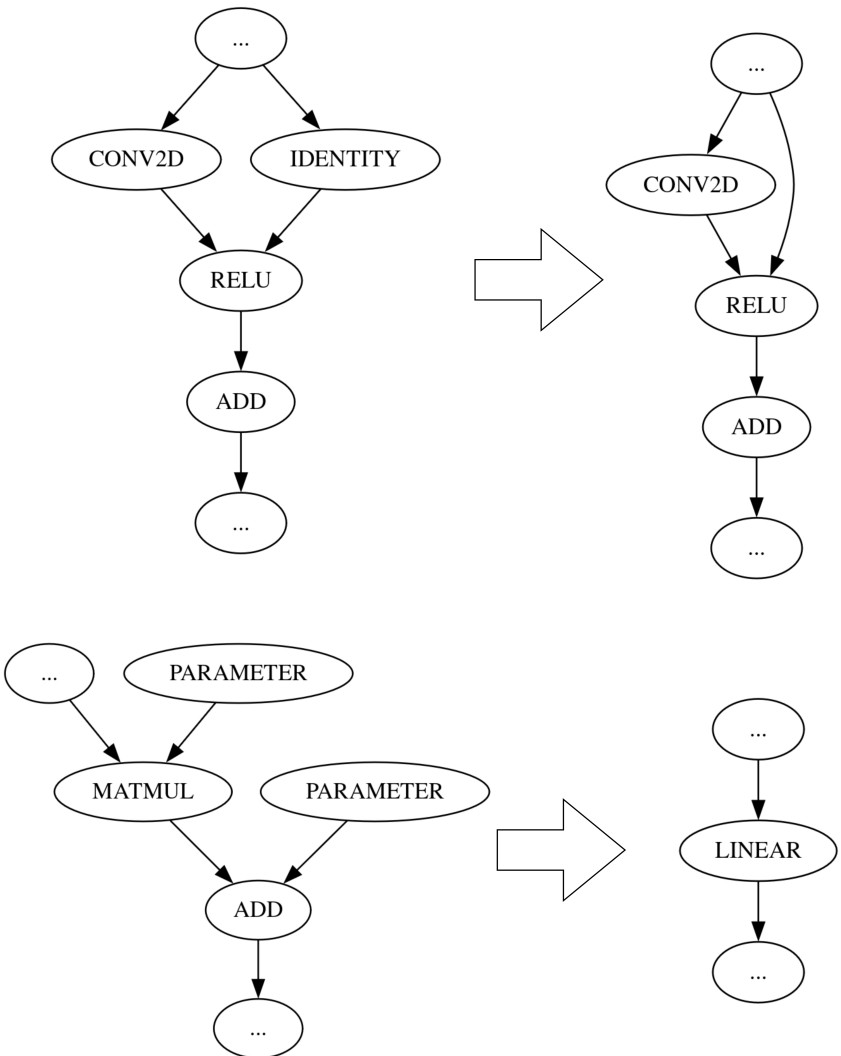

Figure 8: Visualisation of different graph optimisation steps.

Table 12: Zero-shot transfer learning from NAS-Bench-101 to NAS-Bench-201 using different LM backbone. Avg. Kendall's $\tau$ over 5 random seeds is reported.

| Model | Model Size | NB101 $\rightarrow$ NB201 | | | |
|---|---|---|---|---|---|
| | | 200 | 1000 | 5000 | 50000 |
| ModernBERT-base | 150M | 0.548 | 0.552 | 0.566 | 0.559 |
| ModernBERT-large | 396M | **0.566** | **0.574** | **0.581** | **0.571** |
| Qwen3 | 752M | 0.456 | 0.520 | 0.557 | 0.566 |
| Qwen3 | 2.03B | 0.489 | 0.552 | 0.551 | 0.551 |

```
Conv(1x3x32x32, 128x3x3x3, 128)(dilations=1,kernel_shape=3,pads=1,strides=1) -
-> Value1:1x128x32x32
Relu(Value1) --> Conv(prev, 32x128x1x1, 32)(dilations=1,kernel_shape=1,pads=0,
strides=1) --> Relu(prev) --> MaxPool(prev)(kernel_shape=3,pads=1,strides=1) -
-> MaxPool(prev)(kernel_shape=3,pads=1,strides=1) --
> Value2:1x32x32x32
Concat(Value2, Value2, Value2, Value2) --> Value3:1x128x32x32
Conv(Value3, 32x128x1x1, 32)(dilations=1,kernel_shape=1,pads=0,strides=1) -
-> Relu(prev) --> MaxPool(prev)(kernel_shape=3,pads=1,strides=1) -
-> MaxPool(prev)(kernel_shape=3,pads=1,strides=1) --
> Value4:1x32x32x32
Concat(Value4, Value4, Value4, Value4) --> Value5:1x128x32x32
Conv(Value5, 32x128x1x1, 32)(dilations=1,kernel_shape=1,pads=0,strides=1) -
-> Relu(prev) --> MaxPool(prev)(kernel_shape=3,pads=1,strides=1) -
-> MaxPool(prev)(kernel_shape=3,pads=1,strides=1) --
> Value6:1x32x32x32
Concat(Value6, Value6, Value6, Value6) --> Value7:1x128x32x32
MaxPool(Value7)(kernel_shape=2,pads=0,strides=2) -
-> Conv(prev, 64x128x1x1, 64)(dilations=1,kernel_shape=1,pads=0,strides=1) -
-> Relu(prev) --> MaxPool(prev)(kernel_shape=3,pads=1,strides=1) -
-> MaxPool(prev)(kernel_shape=3,pads=1,strides=1) --
> Value8:1x64x16x16
Concat(Value8, Value8, Value8, Value8) --> Value9:1x256x16x16
Conv(Value9, 64x256x1x1, 64)(dilations=1,kernel_shape=1,pads=0,strides=1) -
-> Relu(prev) --> MaxPool(prev)(kernel_shape=3,pads=1,strides=1) -
-> MaxPool(prev)(kernel_shape=3,pads=1,strides=1) --
> Value10:1x64x16x16
Concat(Value10, Value10, Value10, Value10) --> Value11:1x256x16x16
Conv(Value11, 64x256x1x1, 64)(dilations=1,kernel_shape=1,pads=0,strides=1) -
-> Relu(prev) --> MaxPool(prev)(kernel_shape=3,pads=1,strides=1) -
-> MaxPool(prev)(kernel_shape=3,pads=1,strides=1) --
> Value12:1x64x16x16
Concat(Value12, Value12, Value12, Value12) --> Value13:1x256x16x16
MaxPool(Value13)(kernel_shape=2,pads=0,strides=2) -
-> Conv(prev, 128x256x1x1, 128)(dilations=1,kernel_shape=1,pads=0,strides=1) -
-> Relu(prev) --> MaxPool(prev)(kernel_shape=3,pads=1,strides=1) -
-> MaxPool(prev)(kernel_shape=3,pads=1,strides=1) --
> Value14:1x128x8x8
Concat(Value14, Value14, Value14, Value14) --> Value15:1x512x8x8
Conv(Value15, 128x512x1x1, 128)(dilations=1,kernel_shape=1,pads=0,strides=1) -
-> Relu(prev) --> MaxPool(prev)(kernel_shape=3,pads=1,strides=1) -
-> MaxPool(prev)(kernel_shape=3,pads=1,strides=1) --
> Value16:1x128x8x8
Concat(Value16, Value16, Value16, Value16) --> Value17:1x512x8x8
Conv(Value17, 128x512x1x1, 128)(dilations=1,kernel_shape=1,pads=0,strides=1) -
-> Relu(prev) --> MaxPool(prev)(kernel_shape=3,pads=1,strides=1) -
-> MaxPool(prev)(kernel_shape=3,pads=1,strides=1) --
> Value18:1x128x8x8
Concat(Value18, Value18, Value18, Value18) --> Value19:1x512x8x8
ReduceMean(Value19)(axes=[2,3]) --> Gemm(prev, 10x512, 10) --> Out
```

Figure 9: A simple neural network in the text representation of ONNX-Net.

