# OpenReview forum: "ONNX-Net: Towards Universal Representations and Instant Performance Prediction for Neural Architectures"
_ICLR.cc/2026/Conference — Submitted to ICLR 2026_

### Official Review · Reviewer_a8D2 · 2025-10-28

**Soundness:** 2
**Presentation:** 3
**Contribution:** 2
**Rating:** 4
**Confidence:** 3

**Summary:**

Traditional NAS surrogate models predict architecture performance within a fixed search space, which ties them to a specific architecture representation with constrained topologies. To enable research that spans multiple NAS search spaces and to improve generalization across them, the authors unify architectures from different NAS benchmarks in ONNX format and build ONNX-Bench. To learn from these representations, the authors design ONNX-Net, an LLM-based predictor that treats ONNX files as text. The experiments demonstrate the generalization ability of ONNX-Net, and the ablation studies further support the effectiveness of the surrogate’s design.

**Strengths:**

1. The paper unifies search spaces from different NAS works using ONNX. This allows surrogate models to predict performance across search spaces and provides the community with a valuable dataset.

2. The paper is well structured and easy to follow.
   - It explains how ONNX-Bench is built, shows the similarities and differences between the search spaces, and displays the text form of the ONNX files for easy understanding.
   - It provides ablation experiments for the text encoding and shows how each component contributes to training and prediction.

3. Generalization is a key problem for surrogate models. The authors validate ONNX-Net in three ways:
   - Cross search space in subsection 5.1.
   - Zero-shot transfer in subsection 5.2.
   - Cross dataset in subsection 5.3.
   These experiments collectively demonstrate that ONNX-Net achieves well-generalized performance across different search spaces and datasets.

**Weaknesses:**

1. Subsection 5.1 does not compare with other baselines. It is hard to judge how well the proposed surrogate model is in that setting.

2. Table 3 compares zero-shot results only for models trained on 50k samples from NAS-Bench-101 and evaluated on NAS-Bench-201. Readers may want to see zero-shot comparisons for more search spaces ( such as hNAS-Bench-201, NAS-Bench-301 ), since ONNX-Bench collects many spaces from NAS Benchmarks.

3. While the paper demonstrates strong cross-space and zero-shot results, several potential causes behind the OOD behaviors remain under-analyzed:
   - In the all-but-one search space setting, why do some target spaces show weaker OOD performance? Could this be caused by differences in operator op_type across spaces?

   - The ablation shows that Input information and Parameter information clearly contribute to performance. Does this imply that the model relies mainly on information that is independent of operator names in unseen spaces, such as input shapes or operator parameters?

   - Since different NAS benchmarks have different node number distributions, could OOD prediction performance also be affected by such node scales?

**Questions:**

1. ONNX-Net is evaluated with an ONNX-based representation, while other baselines are evaluated with their own representations. Since both the input representation and the surrogate model architecture differ, can this comparison be regarded as fair and meaningful?

2. In subsection 5.2, the correlation after leaving out NATS-Bench is 0.390, while training on all is 0.788. Does this large gap indicate poor generalization?

3. Are the issues listed in Weaknesses 3 reasonable and important? If they are indeed important, could the authors provide more explanations or discussions on these points?

---

> ### Author Response · Authors · 2025-11-21
> **Response to Reviewer a8D2**
>
> We would like to thank the reviewer for their constructive review. We appreciate the detailed assessment of the paper and the recognition of our paper’s strengths.
>
> ---
>
> We will now address the reviewer's questions:
>
> > Subsection 5.1 does not compare with other baselines.
>
> Existing predictors are tightly coupled to cell-based search spaces. They cannot be straightforwardly applied across heterogeneous spaces in a unified protocol, which is what ONNX-Bench enables for the first time. Our experiment provides a first baseline for this unified, leave-one-space-out setting.
>
> > Readers may want to see zero-shot comparisons for more search spaces.
>
> Per the reviewer’s suggestion, we conducted additional one-to-one zero-shot transfers between einspace and other search spaces under the same protocol as Section 5.2 (Fig. 5). To align with Table 2, we report both Spearman’s ρ / Kendall’s τ for training set sizes 200/1000/5000.
>
> |                  | 200         | 1000        | 5000        |
> |------------------|-------------|-------------|-------------|
> | nb101 → einspace | 0.366/0.264 | 0.287/0.199 | 0.230/0.155 |
> | einspace → nb101 | 0.404/0.310 | 0.468/0.351 | 0.470/0.348 |
> | einspace → nb201 | 0.338/0.242 | 0.280/0.198 | 0.257/0.180 |
>
> As expected from the substantial distribution shift shown in Figures 2–3, einspace differs largely from NAS-Bench-101/201 in node counts, operator vocabularies, and achievable accuracy ranges, zero-shot transfer is limited. This result underscores our motivation for ONNX-Bench: gathering a unified, diverse architecture collection to expose the surrogate to heterogeneous architectural patterns and improve generalization to truly unseen spaces.
>
> > In the all-but-one search space setting, why do some target spaces show weaker OOD performance? Could this be caused by differences in operator op\_type across spaces?" and "In subsection 5.2, the correlation after leaving out NATS-Bench is 0.390, while training on all is 0.788. Does this large gap indicate poor generalization?
>
> We clarify that the NATS-Bench results are from subsection 5.1, not 5.2
>
> For our leave-one-out experiment setting, we observe two search space with weaker OOD performance, einspace and NATS-Bench. For einspace, we believe the reason is the large within-space operational divergence and inter-space divergence (as shown in Fig 2), as well as its uniques node and accuracy distribution (as shown in Fig 3). This can also be reflected by the improved performance for NAS-Bench-101, NATS-Bench and NAS-Bench-301 when einspace is left out of the training data.
>
> In terms of NATS-Bench, it includes highly constrained topologies (e.g., majority of the search space has fixed structure with only architecture size varied). Other search space rarely vary size alone with a fixed topology, so holding NATS-Bench out deprives the surrogate of the supervision about how operator parameters and channel/width choices affect accuracy.
>
> > ONNX-Net is evaluated with an ONNX-based representation, while other baselines are evaluated with their own representations. Since both the input representation and the surrogate model architecture differ, can this comparison be regarded as fair and meaningful?
>
> We believe representation and surrogate architecture are inherently coupled. An LM cannot directly ingest adjacency matrix, and a GNN cannot process textual representations. We compare end-to-end surrogates under the same data (nb101/nb201 models), labels (their reported CIFAR10 accuracy) and splits.
>
> ---
>
> We hope these clarifications address the reviewer's concerns and we would like to respectfully ask the reviewer to reconsider their rating; we welcome any further questions and thank the reviewer again for their thoughtful feedback.

---

### Official Review · Reviewer_zxk1 · 2025-10-28

**Soundness:** 3
**Presentation:** 2
**Contribution:** 2
**Rating:** 4
**Confidence:** 5

**Summary:**

This paper is about designing a generalizable predictor for Neural Architecture Search (NAS) using the Open Neural Network eXchange (ONNX) representation standard, and then using a Large Language Model (LLM) to perform the predictions. The name for this framework is ONNX-Net - consisting of ONNX-Bench, the neural networks in (representation, accuracy) pairs, on CIFAR-10, and ONNX-Net, the LLM-based predictor. ONNX-Net is evaluated on some unseen NAS tasks.

**Strengths:**

The strongest contribution of this paper is representing neural network architectures using the ONNX standard.
This is probably the best method to do so as ONNX is a platform for saving a neural architecture on one device, then deploying on another, e.g., for mobile deployment applications.

Further, the reviewer appreciates the operation distribution calculations shown in section 3, e.g., the JSD calculation and Fig. 3. This provides some necessary insights on the distributions of different search spaces.

Extensive experiments are performed measuring the Kendall's Tau and Spearman Rho across different benchmarks, in the transfer context, and on unseen tasks.

**Weaknesses:**

The first weakness of this work is that it is primarily on CIFAR-10 which is an incredibly worn-out benchmark at this stage and unlikely to be a good representative of how an architecture would perform on a higher-resolution task. For instance, NAS-Bench-201 [1] also consider CIFAR-100 and downsampled ImageNet; TransNAS-Bench [2] consider other tasks besides image classification and also provide macro search space architectures; AIO-P [3] only consider macro search space architectures for high-resolution tasks as well. While this work considers hierarchical search spaces as well as cell-based, it fails to substantially reach beyond CIFAR-10 and low-resolution tasks.

Second, the characterization of GENNAPE [4] is not accurate, since GENNAPE is not limited to cell-based architectures, but uses the same representation as [3] which covers macro-search space architectures for cross-task prediction. The description the authors use in the paper is better suited to CDP [5], which is one of the earliest iterations of a generalizable predictor but also confined to cell-based architectures.

Third, the use of an LLM in this paper to predict performance seems like a large leap but doesn't provide sufficient pay-off, given the results, which while not lackluster, are mostly incremental. The reviewer would note that there have been several advances in low-cost predictor design to take advantage of the graph structure [6, 7] that this paper either does not seem to be aware of or simply discards.

**Questions:**

Two questions:
- Can the authors provide further comparison with flow-based [6] predictor models as well as causal predictor models [7]? This would help to better justify the use of an LLM.
- L038: "Recently, researchers have begun to focus on more expressive search spaces that enable the discovery of more diverse and innovative architectures". There is more work in this field than the authors lead on. Are you able to provide some revised/further commentary/dialogue/work on these efforts?

References:

[1] Dong, Xuanyi, and Yi Yang. "Nas-bench-201: Extending the scope of reproducible neural architecture search." arXiv preprint arXiv:2001.00326 (2020).

[2] Duan, Yawen, et al. "Transnas-bench-101: Improving transferability and generalizability of cross-task neural architecture search." Proceedings of the IEEE/CVF Conference on Computer Vision and Pattern Recognition. 2021.

[3] Mills, Keith G., et al. "Aio-p: Expanding neural performance predictors beyond image classification." Proceedings of the AAAI Conference on Artificial Intelligence. Vol. 37. No. 8. 2023.

[4] Mills, Keith G., et al. "Gennape: Towards generalized neural architecture performance estimators." Proceedings of the AAAI Conference on Artificial Intelligence. Vol. 37. No. 8. 2023.

[5] Liu, Yuqiao, et al. "Bridge the gap between architecture spaces via a cross-domain predictor." Advances in Neural Information Processing Systems 35 (2022): 13355-13366.

[6] Hwang, Dongyeong, et al. "Flowerformer: Empowering neural architecture encoding using a flow-aware graph transformer." Proceedings of the IEEE/CVF Conference on Computer Vision and Pattern Recognition. 2024.

[7] Ji, Han, et al. "CARL: Causality-guided Architecture Representation Learning for an Interpretable Performance Predictor." arXiv preprint arXiv:2506.04001 (2025).

---

> ### Author Response · Authors · 2025-11-21
> **Response to Reviewer zxk1**
>
> We thank the reviewer for their constructive review. We appreciate the recognition of our paper’s strengths and the insightful suggestions given.
>
> ---
>
> We will now address the reviewer's questions:
>
> > The first weakness of this work is that it is primarily on CIFAR-10 which is an incredibly worn-out benchmark at this stage and unlikely to be a good representative of how an architecture would perform on a higher-resolution task.
>
> We agree CIFAR-10 is a saturated benchmark and not representative of all tasks. We chose it primarily because it is the only consistent, universal evaluation available across all search spaces we unify, which lets us train and compare predictor and representations at scale with consistent labels. Re-training 600k models on multiple datasets is unfortunately prohibitively expensive. In the original paper, we include cross-dataset evaluation to 8 UnseenNAS tasks within einspace (Table 4), and we plan to extend ONNX-Bench to include additional evaluations as future work.
>
> > the characterization of GENNAPE is not accurate
>
> We thank the reviewer for pointing this out. We have updated our description of GENNAPE accordingly in the uploaded revised version.
>
> > the use of an LLM in this paper to predict performance seems like a large leap but doesn't provide sufficient pay-off
>
> Unlike most prior surrogates that are trained from scratch, leveraging a pretrained language model gives us a flexible predictor that can exploit prior knowledge about operators and compositions encoded in the text representation. This inductive bias is particularly valuable under distribution shift to new search spaces and in low-data regimes, which we believe explains our stronger zero-shot results with limited training points. Importantly, our choice is not “bigger is better”: we use a compact 396M encoder-only model that, in our ablations, outperforms larger decoder-based LLM on this task, and even the 150M variant remains competitive.
>
> > Can the authors provide further comparison with flow-based predictor models as well as causal predictor models
>
> We appreciate the reviewer for drawing our attention to these two works. To the best of our knowledge, both surrogate models are designed for in-domain prediction, i.e., training and evaluation within the same search space. Our study, by contrast, targets cross-domain prediction across different search spaces, so the original evaluation protocols are not directly comparable. We will attempt to adapt these surrogates to our setting and incorporate them in the camera-ready version.
>
> > "Recently, researchers have begun to focus on more expressive search spaces that enable the discovery of more diverse and innovative architectures". There is more work in this field than the authors lead on. Are you able to provide some revised/further commentary/dialogue/work on these efforts?
>
> We thank the reviewer for this suggestion, we agree that our discussion of expressive search spaces should be broadened. In addition to the spaces already cited, we have added further discussion to the revised version. We would appreciate further pointers if there are any more key references we have missed.
>
> ---
>
> We hope the revisions and clarifications address the reviewer’s concerns; we welcome any further questions or suggestions and kindly ask the reviewer to reconsider the rating while taking the rebuttal and revisions into account. We thank the reviewer again for their time and effort.

---

> ### Comment · Reviewer_zxk1 · 2025-11-26
> **Maintaining score**
>
> The reviewer elects to maintain their original score. Some feedback to the rebuttal:
>
> "In the original paper, we include cross-dataset evaluation to 8 UnseenNAS tasks within einspace (Table 4), and we plan to extend ONNX-Bench to include additional evaluations as future work."
> In the reviewer's honest opinion, this work is incomplete in the absence of such experiments, given the current state of NAS.
>
> "We thank the reviewer for pointing this out. We have updated our description of GENNAPE accordingly in the uploaded revised version."
> These revisions are still incorrect and a mischaracterization, specifically lines 074-077.
>
> The rebuttal commentary surrounding LLMs is still flatly unconvincing.
>
> The comparisons with other specialized NAS predictors mentioned is becoming a necessity given the LLM usage in this paper, and arguably more justified since while those techniques are complex, they are arguably less intensive and less "shoehorn"-like, so to speak, than using an LLM for NAS.
>
> "We thank the reviewer for this suggestion, we agree that our discussion of expressive search spaces should be broadened. In addition to the spaces already cited, we have added further discussion to the revised version. We would appreciate further pointers if there are any more key references we have missed."
> There is a broader literature review the authors should explore on this topic.

---

### Official Review · Reviewer_9D7F · 2025-10-31

**Soundness:** 2
**Presentation:** 3
**Contribution:** 2
**Rating:** 2
**Confidence:** 3

**Summary:**

This paper proposes ONNX-Net, a universal surrogate model for neural architecture performance prediction that operates across diverse NAS search spaces. It builds ONNX-Bench, a unified benchmark of 600k architectures in ONNX format, and converts each network into text for LLM-based prediction.

**Strengths:**

The paper introduces ONNX-Bench, which collects architectures from multiple search spaces into a unified ONNX format. The dataset may benefit further research.
The paper explored the ONNX-to-text encoding method that applies to arbitrary architectures.

**Weaknesses:**

The motivation for using the text encoding method is unclear. The authors should further clarify the differences and advantages of introducing text encoding compared to other possible approaches.
The authors argue that using Python code as an architectural representation could produce nonsensical or syntactically incorrect results. I believe the proposed ONNX approach in this paper faces a similar issue, and the authors may need to provide further clarification on the key difference.
I noticed that the authors report Kendall’s τ for some results (Table 2) but Spearman’s ρ for others (Table 3). They should either include both metrics for completeness or explain why different correlation measures are used.
The zero-shot transfer experiments are only conducted on NAS-Bench-101 and NAS-Bench-201, both of which are cell-based search spaces. The authors should also demonstrate the model’s generalization ability across different types of search spaces.

**Questions:**

In addition to the above, I also have a question: how do the authors view the relationship between the ONNX format and the encoded text? It seems that the ONNX-to-text process is essentially a simplification of the ONNX representation to fit the model’s input length. Therefore, can we consider ONNX merely as an intermediate format, and in fact, directly establish a search-space-to-text representation?

---

> ### Author Response · Authors · 2025-11-21
> **Response to Reviewer 9D7F**
>
> We thank the reviewer for their constructive review. We appreciate the detailed assessment of the paper and the insightful suggestions.
>
> ---
>
> We will now address the reviewer's questions:
>
> > The motivation for using the text encoding method is unclear. The authors should further clarify the differences and advantages of introducing text encoding compared to other possible approaches.
>
> The ONNX-based text encoding lets us represent arbitrary operator types and heterogeneous topologies without redesigning the encoder for every search space. Compared to adjacency-matrix encodings, text encodings naturally scales to variable model size, and can include any additional information as desired.
>
> Python code encodings on the other hand, are highly non-canonical: across different deep learning libraries (e.g., PyTorch, TensorFlow, JAX), the same architecture can be implemented with very different code. Even within a single framework, the identical model can be expressed in many syntactically distinct ways (e.g., varying module organization, helper functions, or class structures), creating a many-to-one mapping from Python code to the underlying model. By contrast, ONNX yields far fewer distinct encodings per model. Based on these two observations, we believe that ONNX representations are substantially more sample-efficient than Python code representations, because the model doesn's have to learn invariances to a large variety of superficial code-level differences.
>
> > I noticed that the authors report Kendall’s τ for some results (Table 2) but Spearman’s ρ for others (Table 3). They should either include both metrics for completeness or explain why different correlation measures are used.
>
> - Zero-shot comparisons: We reported Spearman’s ρ in zero-shot NAS-Bench-101 → NAS-Bench-201 to directly compare with FLAN and GENNAPE, which use Spearman’s ρ.
> - Other settings: We used Kendall’s τ elsewhere because it is more robust and less sensitive to large errors in the tails.
>
> We appreciate that this inconsistency makes it somewhat confusing and harder for future benchmarking, so we have include the missing Kendall’s τ numbers in the appendix for completeness.
>
> > The authors should also demonstrate the model’s generalization ability across different types of search spaces.
>
> Per the reviewer’s suggestion, we conducted additional one-to-one zero-shot transfers between einspace and other search spaces under the same protocol as Section 5.2 (Fig. 5). To align with Table 2, we report both Spearman’s ρ / Kendall’s τ for training set sizes 200/1000/5000.
>
> |                  | 200         | 1000        | 5000        |
> |------------------|-------------|-------------|-------------|
> | nb101 → einspace | 0.366/0.264 | 0.287/0.199 | 0.230/0.155 |
> | einspace → nb101 | 0.404/0.310 | 0.468/0.351 | 0.470/0.348 |
> | einspace → nb201 | 0.338/0.242 | 0.280/0.198 | 0.257/0.180 |
>
> As expected from the substantial distribution shift shown in Figures 2–3, einspace differs largely from NAS-Bench-101/201 in node counts, operator vocabularies, and achievable accuracy ranges, zero-shot transfer is limited. This result underscores our motivation for ONNX-Bench: gathering a unified, diverse architecture collection to expose the surrogate to heterogeneous architectural patterns and improve generalization to truly unseen spaces.
>
> > how do the authors view the relationship between the ONNX format and the encoded text? It seems that the ONNX-to-text process is essentially a simplification of the ONNX representation to fit the model’s input length. Therefore, can we consider ONNX merely as an intermediate format, and in fact, directly establish a search-space-to-text representation?
>
> We agree that one could design a search-space-to-text pipeline per space. Our choice of ONNX is because of its unification and forward compatibility. Using ONNX as the representation decouples the surrogate from any specific search space. Any current or future space that can export to ONNX is immediately supported, no new representation needs to be created. We are currently exploring how other formats like NNEF can be used as an alternative.
>
> Our ONNX-to-text process is necessary for the representation to be useful for performance prediction. We build on the universal ONNX representation format but make several important changes to improve performance by condensing the information and removing redundancy. The impact of these changes are shown in the ablation study in Table 5: adding inputs to the Base encoding yields the largest gain in performance, Output and parameter information provide further improvements.
>
> ---
>
> We hope these clarifications address the concerns raised in the review; we welcome any further questions or suggestions and we would kindly ask the reviewer to adjust their review rating taking the rebuttal into account. We thank the reviewer again for their time and effort.

---

### Official Review · Reviewer_JBju · 2025-11-03

**Soundness:** 2
**Presentation:** 2
**Contribution:** 1
**Rating:** 0
**Confidence:** 3

**Summary:**

This work mainly accomplished the representation of mainstream neural network architectures using the ONNX file format, which can be used for NAS research. It has more engineering value and lacks research innovation. In this standardisation process, the technical work also lacks sufficient validation. I suggest the authors to focus just one point, ONNX format or LLM refining on ONNX, with more in-depth research.

**Strengths:**

This work explores the possibility of using ONNX for the unified conversion of network architectures, which provides some inspiration for subsequent research.

**Weaknesses:**

1. The focus of this work is not clear enough. Specifically, is the theme of this paper the unified handling of network representations in the ONNX file format, or is it verifying LLM performance based on this? In either case, the research content is insufficient.

2. For work involving the design and release of a unified representation format, the key point should be that the unified format does not alter the performance of existing models. This is essential to verify the effectiveness and reliability of a compromise unified representation format. However, this paper indicates that the performance of the proxy model changes at this point, which seems abnormal. It is recommended that the authors consider comparing the performance of the same proxy model prediction method under the proposed ONNX format and the original format, and then further demonstrate it.

**Questions:**

I have one big concern. The author claims that 'a surrogate model using the novel text-based encoding trained on ONNX-Bench achieves competitive performance, especially for zero-shot transferability with minimal pretraining.' The question here is, compared with existing work, ONNX-NET only differs in file format or network representation, so why does it lead to model performance improvement? If asked further, is it a general performance improvement or mainly targeted at zero-shot? In fact, I doubt this conclusion.

---

> ### Author Response · Authors · 2025-11-21
> **Response to Reviewer JBju**
>
> We thank the reviewer for their comments. However, we are concerned the reviewer has not fully understood our work, and that the strong reject is unwarranted. We restate the purpose and scope of our work below for their reconsideration.
>
> Our paper is a representation-learning contribution motivated by ICLR's core themes. We propose a universal, search-space-agnostic representation of architectures that enables predicting their performances regardless of how the architecture was defined; in a PyTorch, Tensorflow, Jax framework, or from a collection of diverse NAS search spaces. This representation also allows us to easily transfer knowledge across search spaces. To make this possible we make two tightly-coupled contributions:
>
> - ONNX-Bench: a unified, diverse resource of >600k {architecture, accuracy} pairs in a common format, necessary to train and evaluate cross‑space predictors fairly.
> - ONNX-Net: a general ONNX-to-text encoding that preserves topology and operator‑level details, paired with an LLM performance predictor model to demonstrate the utility of the representation.
>
> The representation and the surrogate model are not two disconnected topics: ONNX-Net is a representation that generalises performance prediction across spaces, and ONNX-Bench is a necessary benchmark for evaluating this.
>
> ---
>
> We will now address the reviewer's questions:
>
> > For work involving the design and release of a unified representation format, the key point should be that the unified format does not alter the performance of existing models.
>
> We agree that a unified representation must not alter the accuracy of the underlying trained models and our proposed approach does not. We do not re-train or re-evaluate models from exported ONNX; The evaluated accuracies (e.g., CIFAR-10) we reported are ground truth from a consistent training pipeline, and ONNX is used only as an interchange format to represent architecture graphs. Any graph "simplification" we perform for encoding is algebraically equivalent (e.g., MatMul+Add → Linear/Gemm), and is used to shorten the text for LLM context limits and to make it more efficient. Thus, there is no change to the original trained model performance; The numbers we predict against are unchanged. The variation the reviewer notes is the surrogate prediction accuracy (i.e., how well a predictor ranks architectures), which is expected to depend on the information and inductive bias of the encoding.
>
> > It is recommended that the authors consider comparing the performance of the same proxy model prediction method under the proposed ONNX format and the original format, and then further demonstrate it.
>
> Thanks for this suggestion. We assume by comparing proposed ONNX format and the original format, the reviewer is referring to our proposed text encoding and the raw ONNX files. The raw ONNX file is binary, and the only readily available textual form via onnx.printer.to\_text has verbose identifiers and metadata (e.g., very long tensor names like /backbone/inner\_fn.0/first\_fn/prerouting\_fn/unfold/Add\_output\_0…), which dominates the token budget. For einspace models, the average tokenized length of this raw serialization is 133,045 tokens, compared with 5,649 for our proposed text encoding, placing the raw form well beyond the context windows of the LMs used in this work and making a fair comparison infeasible without truncation (which would confound the results).
>
> > The question here is, compared with existing work, ONNX-NET only differs in file format or network representation, so why does it lead to model performance improvement?
>
> Representation determines what information is available to a predictor and the inductive biases it can exploit. Prior cross-space surrogates often rely on adjacency matrix encodings and are insensitive to operator parameters; different networks can share the same adjacency but differ in kernel size, stride, padding, etc. In our work, we build on the universal ONNX representation format but make several important changes to improve our performance prediction. We show how these changes lead to model performance improvement in our ablations (Table 5): adding inputs to the Base encoding yields the largest gain in performance, Output and parameter information provide further improvements.
>
> We hope our responses have addressed the reviewer's main concerns, and we would greatly appreciate a reconsideration of our submission.

---

### Author Response · Authors · 2025-11-21
**Acknowledgment to all Reviewers**

We appreciate the reviewers for their insightful suggestions and feedback. We have uploaded a revised version of our paper with changes highlighted in blue. Below are the specific changes we made:

- Added a detailed explanation of performance predictors.
- Added further discussion comparing Python code and text representations.
- Added Kendall's τ to tables which previously just had Spearman’s ρ avaliable.
- Additional zero-shot experiments along with discussion of the results.
- Updated discussion regarding GENNAPE[1].
- Additional discussion about representations for flow-based[2] and causal-based[3] predictors.
- Additional discussion about expressive search spaces.

[1] Mills, Keith G., et al. "Gennape: Towards generalized neural architecture performance estimators." Proceedings of the AAAI Conference on Artificial Intelligence. Vol. 37. No. 8. 2023.
[2] Hwang, Dongyeong, et al. "Flowerformer: Empowering neural architecture encoding using a flow-aware graph transformer." Proceedings of the IEEE/CVF Conference on Computer Vision and Pattern Recognition. 2024.
[3] Ji, Han, et al. "CARL: Causality-guided Architecture Representation Learning for an Interpretable Performance Predictor." arXiv preprint arXiv:2506.04001 (2025).

---

### Meta-Review · Area_Chair_y5Yh · 2026-01-07

**Summary:**

The paper focuses on the important problem of Neural Architecture Search and proposes a benchmark, ONNX, with ca. 600k architectures converted into a common format. The authors demonstrate that the text-based encoding of neural architectures is flexible in describing the characteristics of the considered architectures.

The reviewers expressed important concerns about the novelty and the research depth of the paper, as well as about the motivation for using text descriptions. They also had questions on the experimental scope, the related work accuracy, and the analysis.

I believe the paper cannot be accepted in the current form, and authors are invited to carefully consider the recommendations of the reviewers.

**Reviewer Concerns:**

Reviewer JBju was categorical about rejecting the paper, and the others expressed a firm stance in rejecting the paper.

**Reviewer Scores:**

Consensus is unanimous to reject the paper.

---

### Decision · Program_Chairs · 2026-01-26

Reject